# Intervening on Global Emergencies: The Value of Human Interactions for People’s Health

**DOI:** 10.3390/bs13090735

**Published:** 2023-09-02

**Authors:** Gian Piero Turchi, Davide Bassi, Marco Cavarzan, Teresa Camellini, Christian Moro, Luisa Orrù

**Affiliations:** Philosophy, Sociology, Education and Applied Psychology Department, University of Padova, 35139 Padua, Italy; bassidavide94@gmail.com (D.B.); cavarzanmarco@gmail.com (M.C.); teresa.camellini95@gmail.com (T.C.); christian.moro@unipd.it (C.M.); luisa.orru@unipd.it (L.O.)

**Keywords:** health, emergency, social cohesion, COVID-19, text analysis

## Abstract

Literature about global emergencies and their impact on people’s health underlines the need to improve the social cohesion of human community and the availability of tools to support people and foster community interactions. This paper illustrates research aimed at describing and measuring human interactions in the Veneto community and its changing during the COVID-19 pandemic. 50,000 text occurrences from social media and newspapers about these topics were analyzed between December 2021 and January 2022. People present themselves as members of different teams, pursuing conflicting aims, and attributing the decision-making responsibility of emergencies management exclusively to governments, without considering themselves as active parts of the community. This delegation process on citizens’ behalf can affect their health: by taking minor responsibility in handling the repercussions of these events on the community and by arguing over the most effective way to deal with them, they risk freezing and waiting for action by third parties, thus leaving mutual interactions and the promotion of their own health at a standstill. Local institutions can use these data to shape prevention policies to manage the community’s emergencies and use them as opportunities to promote public health.

## 1. Introduction

Emergency management has been a deeply argued topic in multiple contexts such as terrorism [1,2]), natural disasters [3,4] and public health [5,6,7]. Public health emergency management has raised as a specific field of practice, since the health impact of infectious diseases, environmental catastrophes, and conflicts in recent years have become increasingly relevant from the perspective of strengthening public health systems and protecting communities from naturally occurring and human-caused threats [8]. The term “public health emergency” (PHE) has been brought back into the public eye by the COVID-19 pandemic [9]—a public health emergency so impactful, even in the scientific community, that the WHO database held 742,202 papers on the subject at the end of 2022 [10]. Among these emergencies, this paper considers as a specific case the spread of the COVID-19 in February 2020. This emergency has been selected not only for its impact on the entire globe during the last three years, but also for the prominence it has had in the media and socials in the Veneto region from its start and throughout its course. This was observed by the Hyperion Observatory, the subject of the research proposal of the paper; in fact, the COVID-19 pandemic has been widely mentioned and debated by the Veneto region population in the media and digitally, with more than 50,000 texts a week.

### 1.1. The Emergency Case Selected

The COVID-19 pandemic has proved the necessity for the human community to act together in the pursuit of a common goal: the reduction of the spread of the contagion.

The COVID-19 pandemic has been characterized by several aspects—among others, healthcare workers playing a central role in the management of the sanitary part of the pandemic [11,12] and the increase in psychological-related diseases affecting a multitude of people throughout the human community [13]. Furthermore, many have suffered the loss of loved ones, work, and financial security in a pandemic that counts almost 768 million confirmed cases and 6.94 million deaths globally, and almost 11 million confirmed cases and 145,334 deaths in just the Italian territory, at the time of writing [14]. Ref. [15] reports that globally, in the pre-vaccination period, the median infection fatality rate (IFR) was estimated to be 0.034% for <60 years people and 0.095% for <70 years; in Italy, the rate was, respectively, between 0.10–0.15% and 0.3–0.4%. As of 1 January 2021, in Italy the age-standardized IFR was 0.442% [16].

The complexity of this emergency and its specific criticalities can generate an impact on public health, economics, and people’s mental health on the rest of the world [17,18,19]. Ref. [20]’s study presented some psychosocial effects of the COVID-19 pandemic among different age groups. Young adults had higher scores in preoccupation and change of habits linked to COVID-19, while older adults were least worried and expressed less fear [20]. During the early days of the COVID-19 pandemic, people experienced relevant psychological distress, with symptoms of anxiety, depression, and PTSD observed consistently across the globe [21]. Overall, these behavioral and psychological effects [22,23,24,25,26] plus stress [13,27,28], fear of contagion, death [29,30], and social isolation [31] were the main concerns. In addition, a disinformation campaign was conducted to disrupt the healthcare infrastructures [32].

It is clear that emergencies involve the whole community and require an accountability to each human species’ member, thus making it necessary to consider what is happening with social cohesion [7]. In fact, from psychological and behavioral perspectives, social cohesion can help reduce risky behaviors, diffuse health information [33], and provide social support, which, in turn, can buffer the negative impact of stress [33,34,35,36] and improve health outcomes [37]. The literature offers plenty of definitions for social cohesion [35,36,38,39] that focus on different aspects of the construct. In many cases, it has been shaped as an index composed of several elements, such as social relations, gender equity, social inclusion, and faith [36,38]. Ref. [33]’s study found that three dimensions of social cohesion (social inclusion, social capital, and social diversity) were significantly associated with individual-level self-rated health. Also, social integration is considered a relevant protective factor for the health and well-being of adults [40]: studies in refs. [41,42] pointed out that social integration is, in fact, positively related to self-rated health and satisfaction with life. Another dimension of social cohesion, orientation towards the common good, may also facilitate co-operative behaviors through which communities achieve safety [43] or even influence public health policies [44]. Regarding these last points, in the last years, increasing public administration, starting with European policies and ending with national and regional measures, took social cohesion as a main goal, focusing on reduction of economic and social disparities [45].

Since the 1980s, the European Union considered social cohesion as a fundamental element of its political activity, involving the construct in social policies definition despite the lack of a unique definition of the construct inside EU documents. Social cohesion studies conducted in the Italian territory considered the construct, above all, in terms of immigration and cultural minorities [46,47], impact on the economic side [48], and disaster recovery assets as a main tool in crisis management [49]. Despite this plurality of definitions, social cohesion has not been considered yet as a tool that can help manage the emergency framework that the COVID-19 spread has generated with an eye to safeguarding and promoting people’s health. In fact, during the emergency, the scientific community has made a huge contribution by sharing detailed data strictly inherently in the sanitary sphere (such as daily number of infections, hospitalizations, deaths, and swabs conducted) but missing out on providing any data that describe how the human community was handling the emergency from the interactive and circular health points of view.

### 1.2. The Research Proposal

On the basis of the considerations reported above, the aim of this paper is to present a contribution to the observation of the social cohesion among citizens during this community emergency. More precisely, we assumed that is necessary to consider how the community refers to them in a discursive way, in order to obtain a better understanding and, therefore, a better management of the effects this emergency can have on people’s health. The literature refers to a public health emergency (PHE) as a “sudden occurrence of major infectious disease that causes serious damage to public health, disease of unknown cause in groups, major food or occupational poisoning, or other events that seriously affect public health, which has the characteristics of suddenness, publicity, urgency, and seriousness” [9,50]. In line with, and adding to, that definition, in this study, we consider emergency as “discursive configuration that is triggered by an event (which may be environmental, community, biographical, etc.) related to a change for which the interactive arrangement of the affected community may be altered to the extent that its health and social cohesion, as well as its maintenance/development perspective, are threatened” [51]. Both the definitions outlined account for the fact that the process of generation of a public health emergency, in its broadest sense, is the same. In fact, regardless of the peculiar typology of events and effects (contents, different from each other), all emergencies originate from a sudden event that modifies, in a more or less influential way, the community arrangement involved (process). So, the scope of the paper is to highlight the impact that COVID-19 pandemic has had on community social cohesion and, in turn, on the health of citizens.

In this study, social cohesion is considered as the “whole of the modalities, at discourse level, configuring (designing) realities that concur to the shared management among the community members of the key aspects anticipated, thanks to common goals” [7,51]. In other words, social cohesion stands as the interactive contribution, possible for every citizen, towards a common goal which, ultimately, is people’s health. Thus, health itself is founded on the interaction between the members of the community and can be fostered by social cohesion [7]. According to this definition, every citizen interaction can be observed from a social cohesion point of view. In fact, each discursive production generated by a member of the human species contributes to social cohesion to some degree through the use of natural language, and, in turn, to health. On the highest level of social cohesion, we have goal-oriented (cohesive) discursive productions, which promote a shared management of critical issues, and the pursuit of a shared aim: in the case of COVID-19, to reduce virus infections. On the other extreme of the continuum, at the lowest level of social cohesion, we have productions moving towards personal and implicit goals, thus promoting social fragmentation and reducing the community’s health.

Addressing the observation of discursive interactions requires the adoption of the principle of uncertainty [52]: therefore, moving from a mechanistic paradigm, based on a causal link between factors and prevision, to an interactive one, based on uncertainty of the results and anticipation of the possible outcomes. This paradigmatic shift, in turn, allows the management of interactions generated by all the people belonging to the human community. Moreover, the principle of uncertainty shows that the forecast of emergency scenarios like the above-mentioned COVID-19 pandemic, as well as the extent of its effects in terms of health and interactions, is not possible. Consequently, having discursive data of the mass of the interactions generated by the people who contribute to the narrative configuration of the pandemic allows us to have a measurement index (social cohesion index) of the interactions of the citizens. Through this index, the observation of the objective to which people’s interactions are directed becomes possible, whether it is the needs of the individual (i.e., for the COVID-19 emergency, “I need to see my friends, so I’m going to go visit them in spite of the restrictions”) or the aim that unites the members of the community. These data (the social cohesion index) allow for a comprehensive understanding of the emergency, allowing the directing of health and social policies in order to enhance the contributions of the citizens, and, therefore, a better management of the emergency [7]. In order to make these data available, a social cohesion observatory was established at the University of Padova, known as the Hyperion Observatory (Osservatorio Hyperion, in Italian). The aim of this observatory is to offer a description and measurement of how and how much social cohesion changes in the community of the Veneto region, northern Italy, during the emergency, and how these changes influence the health of the citizens. The observatory gathers accounts produced by people concerning the management of the emergency. These accounts, then, enable the identification of the interactive elements that can be provided to citizens and other major players constituting the community, and can be used to promote a cohesive management of the emergency. How does a population interact during a pandemic? How does a population describe what has happening during the emergency? How much do they share responsibility to reach a common goal? Data obtained by the Hyperion Observatory provide an answer to these questions, granting information useful for a comprehensive understanding of the emergency’s interactive framework, and, therefore, a greater chance to manage it, in pursuit of the people’s and the community’s health. Moreover, these data ensure the anticipation of elements and situations that could happen in the future interactive framework of the community. Weekly results can be utilized by those in institutional roles, such as the president of the region and mayors, as a litmus test of the accounts of the citizens: the availability of the topics of interest of the community and how citizens describe and use what is happening during an emergency can be useful tools to direct choices during decision making. Data can also be used by news organizations and reporters to support their publications with methodologically well-founded elements, like some of Veneto’s newspapers have been doing since the COVID-19 pandemic. Actually, thanks to the work conducted in these two years, the Hyperion Observatory counts more than 40 news articles in local and national newspapers.

In the next paragraphs, an overview of the theoretical–methodological framework adopted by the Hyperion Observatory will be given, along with a description of the observatory’s core components, methodological praxis, results obtained, and future perspective.

## 2. Materials and Methods

This proposal is grounded within the narrativistic paradigm and adopts dialogic science as its theoretical framework [7,53,54]. The methodology used is the methodology for the analysis of computerized textual data (MADIT) [53,54,55]. Within the narrativistic paradigm, reality is intended as generated by the use of natural language and can be considered as a constantly changing configuration (i.e., narrative framework) of all the narrations produced by individuals. According to dialogic science, natural language has been formalized in 24 discursive repertories (DRs) [7,53,54] which describe the method language is used to generate the reality of sense. DRs are organized in the semi-radial table of the discursive repertories (for details, check [7]) and they are divided into three typologies: generative (GR), stabilization (SR), and hybrid (HR). The first category includes those discursive modalities that promote a change from the current status of things, contributing, for example, to the generation of a reality in which the individual describes an aspect of the emergency and generates proposals for its management with an eye to public and people’s health, allowing it to be shared among other community members [7]. Stabilization ones are discursive modalities that generate a stable and immutable reality of sense, supporting, for example, the creation and maintenance of the status of things, not allowing the sharing of the same account of the emergency. Lastly, hybrid modalities are discursive modalities that can have both generative or stabilization valence, depending on the repertories to which they link in their use. As reported in a previous study [7], this formalization allows for a measure of the discursive modalities that builds the configuration of reality of sense by the contribution of different members of the community. As stated above, each discursive production, generated by the use of natural language by a member of the human species, gives a contribution in terms of social cohesion (if oriented to the common and explicit goal, instead of personal and implicit aims).

The social cohesion construct, according to its definition, has been operationalized in two dimensions: anticipation of future scenarios (variable x) and shared management (variable y) [7]. The definition of social cohesion considers the common goal as a reference that guides the anticipation of critical aspects and the shared management of the emergency. Given a common goal, we observe the modalities of pursuing it rather than the efficacy; variables x and y represent two elements that describe how the community moves (in terms of interactions between individuals) and not the goal obtained. Each DR has its own properties and contributes to generate a peculiar reality of sense (see Attached Material in [7]) by giving a contribution to one of the two variables (see Table 1 and Table 2). The degree of contribution of each DR is represented by its related dialogic weight (dW). Thanks to the interactions between the DRs that generate the reality of sense configuration and to the variables x and y, a social cohesion index is available. While the use of some DRs contributes to the pursuit of the common goal, the use of others less so. The DRs, by interacting with each other, render a degree of cohesive contribution that varies according to their use in pursuit of the common goal (which varies according to the emergency). If the DRs are aimed at pursuing different goals to the one set, then the degree of cohesion is assumed to decrease [7].

The text produced by the citizens generates a higher or lower level of social cohesion, according to the DRs used. For example, using a stabilization DR to comment on a publication reporting epidemiological data about the COVID-19 pandemic (such as making a judgment or certifying reality), a discursive configuration is produced, different to the one that would have been generated by the use of a generative DR. Delving into the details of this distinction, generative DRs use third-party references to generate the description of multiple and possible future scenarios that can develop from the present one (i.e., referring to the COVID-19 pandemic, “I am a nurse, I live in Padua, and after work I take swabs for the neighborhood where I live”). This allows the whole community to utilize the elements described to manage the emergency. Stabilization repertories, on the other hand, are used to define a narrative configuration of reality that is fixed, and is not going to change in the future (i.e., “COVID-19 is not real, we are not in danger”). This is very likely to generate a conflict with accounts that convey a different discursive reality (i.e., “After the COVID we need to think of a new way of thinking and structuring the future, both economic and above all social, remembering that nothing is immutable”). The social cohesion index represents the spectrum of different combinations of DRs used in discursive configurations, assuming values from 0 to 20.

Thus, the Hyperion Observatory gathers data from the texts produced by Veneto citizens on online platforms, analyses them through MADIT and the social cohesion index, and makes the results available for the community through a weekly report. In order to measure the social cohesion degree and monitor its weekly variations in the Veneto region, since April 2020, the Hyperion Observatory has selected a sample of about 50,000 textual occurrences every 7 days that refers to the discourses published on social networks and newspapers by citizens and people in political roles, analyzing the language used in them. This process implies different steps and involves different roles which will be described according to the order in which they take place. First of all, the text is gathered by an équipe of 22 text analysts and supervisors using 8 keywords that allow for the retrieval of semantically relevant texts for the topic under investigation. The 8 keywords are chosen from a database resulting from the collection of newspaper articles (of 5000 words): Sketch Engine’s algorithm [56,57] identifies the most frequent words which are then evaluated by the analysts and supervisors, and selected following a semantic classification of relevance to and rate of occurrence with the topic. The 8 keywords are (in Italian): ‘Emergenza COVID-19’, ‘Pandemia COVID-19’, ‘COVID-19 Veneto’, ‘Virus Veneto’, ‘Contagi Veneto’, ‘Virus COVID-19’, ‘Emergenza sanitaria Veneto’, and ‘Crisi sanitaria’. In order to provide a varied sample of text, the sources accessed are those shown in Table 3:

Considering the wide number of texts available from the sources mentioned, the criteria for selecting data are the following:time: texts must have been published from day 1 to day 7 of analysis (in order to collect the discursive configuration of the week);content: the topic of texts must concern the emergency;author: citizens, president and aldermen of the Veneto region, the mayor of the Veneto region province cities, journalists;geographical coverage: all the provinces of the Veneto region.

Concerning the number of occurrences collected weekly, approximately 2500 occurrences are acquired for both Facebook and Twitter–Instagram sources, and 5000 occurrences for both Google and newspaper sources per day, for an amount of 52,500/week. Text is analyzed through MADIT. In order to provide methodologically well-funded data, Hyperion Observatory text analysts receive monthly supervision sessions, where criticalities and doubts which emerged throughout the analysis procedure are managed with the help of senior supervisors and the scientific supervisor. Alongside the analysis, the observatory also produces a content analysis in order to identify which major topics are expressed by the citizens, by categorizing into threads the subjects around which people’s accounts are focused (e.g., school reopening’s criticalities and citizens’ opinions on vaccines). Every week, the Hyperion Observatory allowed for the observation of the social cohesion of Veneto’s community, tracing its fluctuation with a report focused on COVID-19, until 6 January 2022.

## 3. Results

After a year and a half of analysis and about 2 million lexical occurrences, Hyperion observed more than 35,000 discursive repertories and has offered to the community a volume of 90 weekly reports. Each report represents a snapshot of the discursive modalities used by Veneto region citizens to narrate the emergency, providing an indication about the social cohesion of the community and highlighting the major subjects which have been the topics of interest of the people’s accounts (see Appendix A for an example report for COVID-19). As another example of the results offered by the Hyperion Observatory, the following graphic represents the social cohesion trend in Veneto regarding COVID-19, from 27 April 2020 to 6 January 2022 (see Figure 1). The colored bands represent the phases of the pandemic period in Italy over weeks (*x*-axis), while the fluctuating line shows the change in the degree of social cohesion (*y*-axis).

From Figure 1 it can be seen that the highest degrees of social cohesion (>12) were measured in conjunction with the first phase of the pandemic, corresponding to the lockdown period. These data find confirmation on a pragmatic level and attestation on a national scale in how political–administrative roles and citizens took action. In fact, with respect to the former, both the President of the Republic and the Prime Minister urged citizens to consider their actions from a community perspective, thinking about the effects their behaviors could have on the nation’s ability to overcome the emergency, and to unite in “a common sense of purpose” through “involvement, sharing, harmony” as a “vital” aspect [58]. The second, citizens, showed their sense of cohesion and community through so-called “balcony performances” [59], occasions in which people of all ages interacted with each other from their respective balconies and/or windows through music, singing, and dancing [59,60]. The subsequent period, on the other hand, saw a sharp decline in the degree of community social cohesion in the Veneto region (the lowest recorded), also in line with what was reported at the national level. In fact, not only did the “balcony performances” stop [60], rather, misalignments between governmental instructions and regional decisions arose and fragmented, mutually conflicting information was offered (including from scientific sources), and fractures opened up between official com munications at the policy, scientific, and dissemination levels, leading to the appearance of fake news and disaggregated local reactions [61]. This reduced the sense of com munal responsibility conveyed to citizens, with an impact at the level of community social cohesion that led to less compliance with sanitary restrictions and, thus, an increase in the spread of contagion [61].

In order to provide an illustration on how the Hyperion Observatory can be used in an emergency scenario (given that it enables analysis of all possible emergencies), one example of a weekly report for the COVID-19 pandemic is analyzed below. The analyzed period (see Figure 2 and Table 4) is:31 December 2021–6 January 2022, for the COVID-19 pandemic, which resulted in a social cohesion index of 10.76.

## 4. Discussion

Regarding the COVID-19 report (see also Appendix A), which considers discursive modalities generated in the week of 31 December–6 January, the social cohesion index value has decreased during the week (from 11.24 to 10.76). The detected configuration allows the Hyperion Observatory to observe this fluctuation, given by the value that citizens attribute to what happens during the emergency.

Actually, the 67.83% of textual data highlights that the people of the Veneto region tend to narrate themselves and their institutions as members of different teams, which pursue conflicting aims (mutually exclusive and incompatible with each other), as one of return to work, and the other based on a political nature. This exacerbates an interactive arrangement whereby each individual acts as “every man for himself”, representative of a lack of trust in government or weak trust and cohesion within communities [62]. The above-mentioned percentage of analyzed data uses this contrast to estimate the goodness of the objective itself: it is judged on not shared and exclusive criteria of justice, and to delegitimize health indications (such as the vaccination campaign and the mandatory PPE for switching between different territorial zones of emergency), as they are considered useless. This would lead to actions similar to the actions which occurred in the first phase of the pandemic, in which several people were investigated and sanctioned for leaving the red areas, illegal hangouts were organized, and local businesses did not respect the restrictions [61].

Furthermore, the analyses show the tendency of citizens to justify their choices on the basis of the opposition of vision and role against the institutions, as decision-makers. This approach is supported and confirmed by negative value connotations (judgments) and comments (in particular, the DRs of comment and judgement). Therefore, the value attributed to the reality of a health emergency is currently highlighted by the way in which distinct aims are attributed to different social roles (doctors, politicians, workers, parents, etc.) precisely by the light of the position they hold in society. In fact, the rhetoric of “the doctor’s goal”, “the interest of the politician”, “what is dear to the parent”, etc. have been traced. If the use of such discursive modalities (of justification, judgement, and comment) continues, the Hyperion Observatory underlines the risk that, with the possible increase in the number of infected people, the divergence of objectives (so, a fractured community where everyone is configured as a bearer of its own interests) can generate narrations of blame and accusation, addressed exclusively to some specific roles (thus, anticipating rhetoric such as “the fault lies with politicians/parents/teachers”). These would be similar to the ones observed during the first half of 2020, where the government was highly criticized for the decisions taken (often fragmented and unclear) and held totally responsible for poorly handling the emergency [61]. In opposition to the feature of social cohesion as orientation towards common good [43,44], this could open to an increase in the shattering of the community with the relapse of conflicts between roles and, thus, a possible reduction of people’s health and increasing the risk of aggravating the sanitary crisis [62]; while pursuing the same objective of reducing the spread of the infection, they can be configured as members of different teams.

Conversely, the more enhanced is the contribution of each citizen to the common goal of reducing the spread of the infection, the more cohesive is the community, and a greater adherence to health standards can be observed (as its implication). In fact, the 32.17% of the discursive productions analyzed by the Hyperion Observatory in the week from 30 December to 6 January highlights the tendency of some citizens to use the contribution of each member of the community (both those offered by citizens and by institutions) as it is useful for the pursuit of the aim of reducing the spread of the infection; this, in the light of considering that every member of the community, by virtue of their roles played in society (worker, politician, parent, etc.), shares the responsibility in pursuing the common goal (this emerges from the use, however limited, of the description DR).

Summarizing, through the report outlined, we observe how during the COVID-19 pandemic, people living in the Veneto region employ a process of almost full delegation to third-party roles. This removes from the hands of the people themselves the responsibility towards the management of the emergency, but also—and above all—towards the effects that this event may have on their health. By waiting for the action of such third parties, the community exposes itself to the risk that the safeguarding of the health and interactions of citizens remains uncertain. In fact, in the case of the COVID-19 pandemic, efforts were mainly allocated on the sanitary level, while the interactive and global-health dimension of citizens was put on the back-burner, thus reducing the possible contribution of citizens themselves in its management. Within this framework, the Hyperion Observatory reports still provide indications for the community and its institutions on critical and virtuous interactive modalities adopted by the citizens. Indeed, the examples in the previous paragraph show the expendability potential of the information offered by the Observatory. In fact, Hyperion is a tool that can be applied in any city, region and country, enabling public policy decisions to be made based on scientific data not only sanitary related. The considerations made so far—which are displayed to the relevant roles through the weekly reports as a simplified and usable data assessing the community’s cohesion degree and describing its impact on people’s overall health with critical issues where action is needed and strengths to leverage—can be included in the health policy-making process, allowing the institutions to manage in anticipation the possible criticalities that could rise among the community. As an example, since communication strategies are a fundamental element for emergency management in terms of planning, response, and recovery [63], these data could be used both by national and local political institutions to convey information and instructions aimed at reducing the spread of the infection and the emergency’s impact on people’s health in a consistent, straightforward, and clear manner—in contrast to what has been done previously (and described above; [61])—so as to promote greater convergence and cohesion toward sanitary and social instructions to follow. At the same time, the highlights on the cohesive modalities adopted by the citizens can be used as resources in the promotion of a cohesive and consistent management of the emergencies. Events like the above-mentioned “balcony performances” are an essential element for social cohesion [64] and a prime example of an interactive arrangement which fostered a sense of belonging and community, based on a common perspective, which helped citizens deal with the potential repercussions of the emergency on global health [59,65,66,67].

Regarding the limitations of the research proposal, we address the time-consuming human work of data analysis for the denomination of discursive repertories in all the weekly gathered texts; currently, this process is performed by 22 experts at analyzing texts through MADIT. To handle this limitation, two proposals are being conducted. First of all, the training of the text analysts: the denomination methodology is taught through ad hoc training courses that aim to increase both the expertise and the efficiency of analyst roles in the process of discursive repertories annotation [68]. The creation of a professional figure with a high degree of specialization is necessary in order to be able to annotate the repertories of a large number of texts (both for public and private organizations). Following this, the second proposal being conducted is the implementation of machine learning and natural language processing in the analysis process, in order to automate the analysis of big data and make the current work faster and more accurate. Due to the collaboration between the Fisppa Department and the Mathematics Departments of Unipd, for which we are thankful, a ML model for the automated denomination of DRs has been developed and is currently undergoing refinement to increase its accuracy. In future developments of the research proposal, further experiments are planned, aimed at comparing such a model with other ML techniques in order to validate what has already been achieved. Moreover, it is intended to extend and, in parallel, to specify the analysis that is currently conducted through other methodologies; first of all, to conduct surveys on particular target groups (e.g., distinguished by location, age, SES, etc.) with respect to specific and differentiated emergency topics. To date, one of these could be the current energy emergency. This will make it possible to observe whether people describe all emergencies in the same way, or, if otherwise, there are specifics for each emergency according to the related characteristics and/or to the peculiar elements of each target group.

## 5. Conclusions

Nowadays, the management of emergencies’ impact on human health is a challenge that needs to be addressed by many communities around the world. An increasing number of studies is being conducted in order to provide scientific and evidence-based methods to face emergencies [69], nevertheless, no national or global data are available regarding how emergencies affect human interactions and how this has an impact on people’s health.

In order to provide a contribution in the analysis of community interactions related to emergency management, in this paper we presented methods and results of the research proposal of the Hyperion Observatory, the first permanent cohesion observatory adopting dialogic science assumptions. Because social cohesion is a pivotal resource for pre-emergency, acute, and post-emergency management [70] and health promotion [7], we showed how, observing people’s discursive modalities in every available text through the discursive repertories, to measure how much community interactions converge on a common goal and a shared management of the public health emergency becomes possible, and we expressed it in a social cohesion index.

Moreover, given the uncertainty that characterizes these global emergencies, anticipation becomes a fundamental and useful competence that can be used both to manage critical issues before they occur and to promote a more cohesive, consistent, efficient, and effective governance of the emergency, with a permanent focus on increasing people’s health. Thus, the Hyperion Observatory will keep monitoring the trend of the social cohesion index, as well as the interactive modalities adopted by the community, in order to provide as much anticipation as possible to share with the Veneto region and Italy’s institutions, and, thus, increase the degree of social cohesion.

## Figures and Tables

**Figure 1 behavsci-13-00735-f001:**
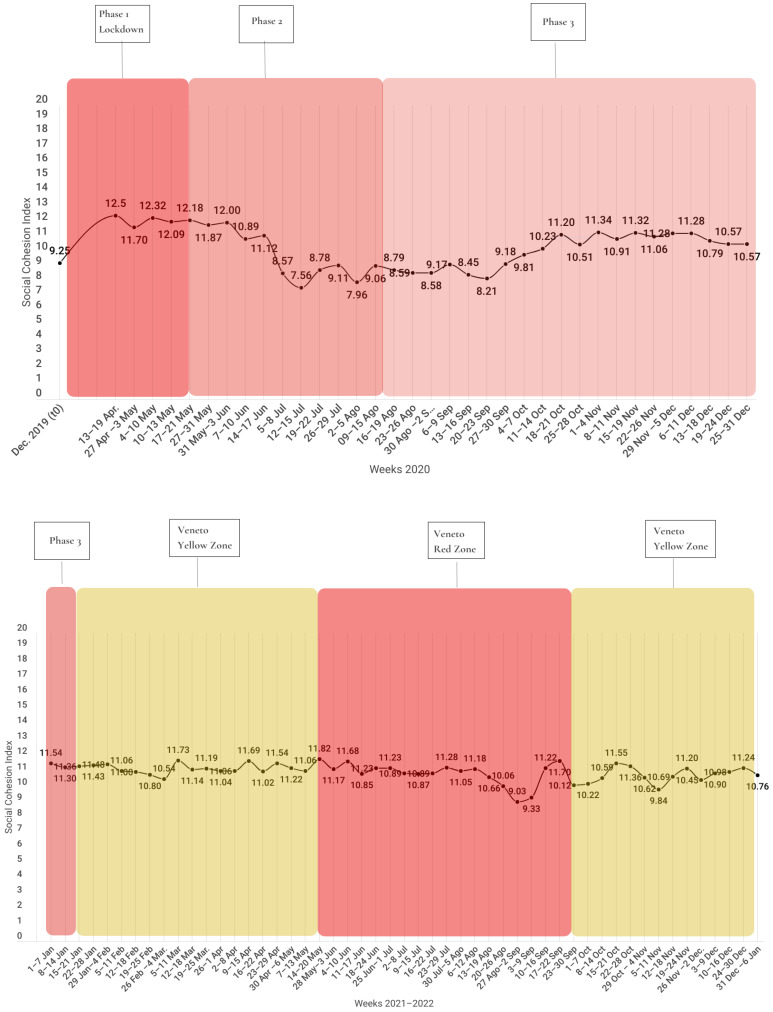
Trend of social cohesion degree in the Veneto region community from April 2020 to January 2022.

**Figure 2 behavsci-13-00735-f002:**
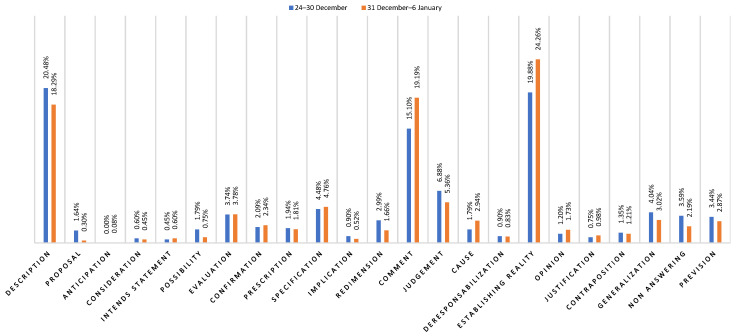
Discursive repertories comparison between 24–30 December and 31 December–6 January on the COVID-19 emergency.

**Table 1 behavsci-13-00735-t001:** DRs pertaining the “anticipation of future scenarios” variable.

DR	dW Contributing to Variable X
Cause of Action	1.74
Prediction	3.48
Confirmation	5.22
Generalization	6.96
Specification	8.7
Implication	10.44
Consideration	12.18
Anticipation	19.14
Description	20

**Table 2 behavsci-13-00735-t002:** DRs pertaining the “shared management” variable.

DR	dW Contributing to Variable Y
Certify Reality	0.87
Contraposition	2.61
Judgement	4.35
Non answer	6.09
Justification	7.83
Delegating to others	9.57
Evaluation	11.31
Opinion	13.05
Declaration of Aim	13.92
Prescription	14.79
Comment	15.66
Reshaping	16.53
Possibility	17.4
Proposal	18.27

**Table 3 behavsci-13-00735-t003:** Sources for textual occurrences retrieval.

Source	Type of Textual Occurrences Retrieved	% of Textual Occurrences Retrieved ^a^	List of Specific Sources Exploited
**Facebook**	Comments produced by the citizens to publications on Facebook pages of the governor of the Veneto region and people in other political roles and from the newspaper’s pages.	25%	- The Veneto region’s president’s page - Regional councilors’ official pages - Profiles of healthcare professionals affiliated with public and private local health institutions
**Twitter**	Text produced by the citizens as tweets directed at (or comments in response to) posts published in Twitter profiles of political figures and from the newspaper pages’ posts.	10%	- The Veneto region’s profile - The Veneto region’s municipalities profiles - The Veneto region’s president’s profile - Profiles of the councilors of the Veneto region and of each municipal council - Profiles of healthcare professionals affiliated with public and private local health institutions
**Instagram**	Text produced by the citizens either as comment to a publication on Instagram pages of the considered people in political roles and from the newspaper pages’ posts.	10%	- The Veneto region’s profile - The Veneto region’s president’s profile - Profiles of healthcare professionals affiliated with public and private local health institutions
**National and** **regional news** **organizations**	Text of the post (including title and subtitle) published on the Google platform by national and regional news organizations.	30%	- Ansa - Ansa Regione Veneto - Antenna Tre - Il Corriere del Veneto - Il Corriere delle Alpi - I Gazetin - Il Gazzettino - Il Resto del Carlino - L’Amico del Popolo - L’Arena - L’Azione - La Difesa del Popolo - La Vita del Popolo - Messaggero Veneto - QDP News Veneto
**Local newspaper** **and dissemination** **websites**	Text of the post (including title and subtitle) published, which contains ‘Veneto region’ and/or the name of a Veneto region province city.	25%	- Il Giornale di Vicenza - Il Mattino di Padova - Il Mestre - Il Padova - Il Treviso - Il Venezia - Il Verona - Il Vicenza - La Nuova di Venezia e Mestre - La Nuova Venezia - La Tribuna di Treviso - Oggi Treviso - Padova Oggi - Rovigo Oggi - Verona News - Verona Oggi - Verona Sera - Vittorio Veneto—Virgilio - Voce di Rovigo - www.epicentro.iss.it (accessed from 1 February 2020)

^a^ The percentage of text occurrences collected from each source was assessed and chosen according to the number of texts relevant to the topic that could actually be found.

**Table 4 behavsci-13-00735-t004:** Distribution of discursive repertories on the COVID-19 emergency (31 December–6 January).

	COVID-19 31 December–6 January	
Discursive Repertory	Frequency	%
Description	242	18.29%
Proposal	4	0.30%
Anticipation	1	0.08%
Consideration	6	0.45%
Declaration of aims	8	0.60%
Possibility	10	0.76%
Evaluation	50	3.78%
Confirmation	31	2.34%
Prescription	24	1.81%
Specification	63	4.76%
Implication	7	0.53%
Reshaping	22	1.66%
Comment	254	19.20%
Judgement	71	5.37%
Cause of action	39	2.95%
Deresponsibility	11	0.83%
Certify Reality	321	24.26%
Opinion	23	1.74%
Justification	13	0.98%
Contraposition	16	1.21%
Generalisation	40	3.02%
Non answer	29	2.19%
Prevision	38	2.87%
TOTAL	1323 ^1^	100%
Social Cohesion Index	10.76	

^1^ The numbers displayed refer to the discursive repertories analyzed, and not to the words collected. The latter reached 100,000 in both surveys.

## Data Availability

Data are contained within the article or Appendix A. The data presented in this study are available in Appendix A.

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
