# Peer review of "Intervening on Global Emergencies: The Value of Human Interactions for People’s Health"

_behavsci, 2023, doi:10.3390/bs13090735_

Round 1
Reviewer 1 Report (New Reviewer)
This research paper deals with how Hyperion Observatory can be used to collect data from the texts produced in social media, news and other websites on public health emergency situation in Veneto region and analyze discursive repertories to provide indication of social cohesion in the community. The paper offers that Hyperion can be applied in any city, region and country to formulate public policy decisions based on the scientific data.
Author Response
We are very grateful to the reviewer for the feedback given to the paper.
Reviewer 2 Report (New Reviewer)
This paper is of interest in analysing the impact of global emergencies on people's health by studying the value of human interactions in a region of Italy during the COVID-19 pandemic. The authors suggest that people narrate themselves as members of different teams, pursue conflicting goals, and attribute responsibility for emergency management decision-making explosively to governments, without considering themselves as an active part of the community. Furthermore, the authors point out that this process of citizen disempowerment can negatively affect people's health.
It is suggested that the authors revise the title of the manuscript as it is too long. It is recommended that it should be no longer than 15 words.
In relation to the introduction and the state of the art, the article presents an interesting and comprehensive review of the scientific literature, primary and secondary scientific sources. Furthermore, the paper succeeds in addressing a more than acceptable number of references and studies on this topic (62), the authors draw on 38 references published between the last three years (2020+2021+2022).
Regarding the method, the authors have carried out a study based on qualitative methods. In this sense, the study is very interesting, well-designed and well-founded. On the other hand, I wonder if it would be possible for the authors to provide any results on the reliability of the analysis carried out.
Finally, the discussion of the results could be enhanced by a review of more current scientific literature.
Author Response
First and foremost, we would like to thank the reviewer for the precious comments and insights given to the paper. Here follow the point-by-point responses:
- This paper is of interest in analysing the impact of global emergencies on people's health by studying the value of human interactions in a region of Italy during the COVID-19 pandemic. The authors suggest that people narrate themselves as members of different teams, pursue conflicting goals, and attribute responsibility for emergency management decision-making explosively to governments, without considering themselves as an active part of the community. Furthermore, the authors point out that this process of citizen disempowerment can negatively affect people's health.
- It is suggested that the authors revise the title of the manuscript as it is too long. It is recommended that it should be no longer than 15 words.
Done. The title has been shortened to 12 words.
- In relation to the introduction and the state of the art, the article presents an interesting and comprehensive review of the scientific literature, primary and secondary scientific sources. Furthermore, the paper succeeds in addressing a more than acceptable number of references and studies on this topic (62), the authors draw on 38 references published between the last three years (2020+2021+2022).
- Regarding the method, the authors have carried out a study based on qualitative methods. In this sense, the study is very interesting, well-designed and well-founded. On the other hand, I wonder if it would be possible for the authors to provide any results on the reliability of the analysis carried out.
In order to offer data to support the validity of the analysis conducted, we have included in the Results section some additional elements and literature references that show how the results obtained by Hyperion in relation to the Veneto Region are in line and consistent with what emerged in the rest of the national territory.
- Finally, the discussion of the results could be enhanced by a review of more current scientific literature.
Done. As suggested, we have added additional and more recent elements from the literature, as well as moved some arguments from the Conclusions section to the Discussion one.
Reviewer 3 Report (New Reviewer)
Thank you for the opportunity to review your manuscript which provides a comprehensive overview of the research proposal related to the Hyperion Observatory's study on social cohesion during the COVID-19 pandemic. It is evident that the proposal aims to explore the impact of the pandemic on community social cohesion and health through analyzing discursive interactions among citizens. The concept of using a Social Cohesion index to measure and understand these interactions is intriguing, as it may offer insights into how individuals respond to the crisis and contribute to a shared goal of managing the emergency.
Abstract: succint, ultiamte conclusion is jsutified in the research, the sentence 'This unaccountability process on citizens’ behalf can affect their health: by not taking responsibility for the repercussions of emergencies on the community to which they belong, and by continuing to give rise to internal disputes over the most effective way to deal with these events, they risk freezing and waiting for action by third parties, thus leaving mutual interactions and the promotion of their own health at a standstill.' although true, is perhaps too strong and unjustified in the manuscript to include in the abstract
Introduction. well written
Material and Methods. comprehensive. It is worth expanding and justifying further around data sources, weighting and how texts were chosen (random, generated algorithm etc).
Results. good representation.
Discussion. although there is some discussion of the results, there needs to be more. what else has the observatory collected to consider a baseline for this data? how do we know that the veneto community is not the outlier, vs the emergency? If there is no baseline, then how do we know that the pandemic is the causative factor and not associative factor? Furthermore, there has not been a discussion around how this data can or is used beyond the generation of a social cohesion index and publishing it so that someone else may use it. This is far too complex for the public to understand, what parts of the government or what niche areas of academic organisations can use this in an impactful way? How has other literature justified and are there similarities with your data?
Conclusion. this is too long and much of it belongs in the discussion. aim to keep the conclusion to 1-2 paragraphs.
some minor grammatical errors
Author Response
First and foremost, we would like to thank the reviewer for the precious comments and insights given to the paper. Here follow the point-by-point responses.
- Abstract: succint, ultiamte conclusion is jsutified in the research, the sentence 'This unaccountability process on citizens’ behalf can affect their health: by not taking responsibility for the repercussions of emergencies on the community to which they belong, and by continuing to give rise to internal disputes over the most effective way to deal with these events, they risk freezing and waiting for action by third parties, thus leaving mutual interactions and the promotion of their own health at a standstill.' although true, is perhaps too strong and unjustified in the manuscript to include in the abstract
Done. As suggested, we changed the sentence mentioned by making it less strong and striking.
- Introduction. well written
- Material and Methods. comprehensive. It is worth expanding and justifying further around data sources, weighting and how texts were chosen (random, generated algorithm etc).
Done. We further specified the sources from which we collected the texts and detailed the retrieval methods, keywords, and percentages of text taken from each source (lines 299-306). This both in the body of the text and with a dedicated table (for readability purposes). - Results. good representation.
- Discussion. although there is some discussion of the results, there needs to be more. what else has the observatory collected to consider a baseline for this data? how do we know that the veneto community is not the outlier, vs the emergency? If there is no baseline, then how do we know that the pandemic is the causative factor and not associative factor? Furthermore, there has not been a discussion around how this data can or is used beyond the generation of a social cohesion index and publishing it so that someone else may use it. This is far too complex for the public to understand, what parts of the government or what niche areas of academic organisations can use this in an impactful way? How has other literature justified and are there similarities with your data?
Since Hyperion was created immediately after the onset of the pandemic, we have no pre-emergence baseline data (neither for the Veneto Region nor for other Italian regions). However, we have included in the Results section some additional elements and literature references that show how the results obtained by Hyperion in relation to the Veneto Region are in line and consistent with what emerged in the rest of the national territory, demonstrating that it is not an outlier (lines 366-.
Moreover, we further specified how the Observatory's data can be used by political institutions for emergency management and response (lines 479-486).
- Conclusion. this is too long and much of it belongs in the discussion. aim to keep the conclusion to 1-2 paragraphs.
Done. As suggested, we moved the majority of the arguments from the Conclusions section to the Discussion one.
This manuscript is a resubmission of an earlier submission. The following is a list of the peer review reports and author responses from that submission.
Round 1
Reviewer 1 Report
It would be essential to emphasize more in the paper that the war conflict is escalating, and the end of the conflict seems to be very far away. Thus, refugees do not have a chance to return to Ukraine. Their number is accumulating, which results in a challenge augmenting day-by-day for host countries. Refugees themselves suffer from a lack of control; without economic potential, they are desperately hopeless (Khullar & Chokshi, 2019).
Social integration is considered a significant protective factor concerning the health and well-being of adults (e.g., Doré et al., 2016). Some research pointed out that social integration is negatively correlated to mental illness and is positively related to self-rated health and satisfaction with life (e.g. Xia & Ma, 2020; Herberholz & Phuntsho, 2018). Perceived stress also significantly mediates the influence of social integration on the dimensions of psychological well-being. Since employment is the potential for enhancing social integration, having an income makes a difference in their mental well-being.
Author Response
We start by thanking the reviewer for the valuable comments. Responses to the specific points follow.
- It would be essential to emphasize more in the paper that the war conflict is escalating, and the end of the conflict seems to be very far away. Thus, refugees do not have a chance to return to Ukraine. Their number is accumulating, which results in a challenge augmenting day-by-day for host countries. Refugees themselves suffer from a lack of control; without economic potential, they are desperately hopeless (Khullar & Chokshi, 2019).
The suggestions and reference have been added (lines 72-73 and 97-99) - Social integration is considered a significant protective factor concerning the health and well-being of adults (e.g., Doré et al., 2016). Some research pointed out that social integration is negatively correlated to mental illness and is positively related to self-rated health and satisfaction with life (e.g. Xia & Ma, 2020; Herberholz & Phuntsho, 2018). Perceived stress also significantly mediates the influence of social integration on the dimensions of psychological well-being. Since employment is the potential for enhancing social integration, having an income makes a difference in their mental well-being.
The relevant suggestions and references have been added (lines 112-115).
Reviewer 2 Report
The submission has a very interesting and worthy academic purpose and goal. However, there are some very problematic aspects related to the research design of the article and to some of the content. In no particular order given, some of the points are listed below.
1) The two cases - COVID-19 and the Ukraine War are very different events in scope and scale. It is very difficult and problematic to try and compare them. COVID-19 was global and Ukraine War is regional in nature, not to mention the scale and absolute number of people affected. Health crisis versus armed conflict is something else to be taken into account, especially given the direct influence and experience of COVID-19 was much more widely experienced than the Ukraine War.
2) The data for COVID-19 affected is problematic and seems to be measured by Case Fatality Rate rather Infection Fatality Rate. For example, a recent freedom of information request in Israel revealed no person aged between 18-49 without a co-morbidity died from the virus.
3) The psychological and cognitive effects of COVID-19 versus Ukraine War are not equal and rather different, not least of which are the direct versus indirect experiences of many people. Hence this shapes an understanding and reaction to the event via perceptions of voluntary versus involuntary risk. People are not greatly increasing drug use and suicides because of the Ukraine War, but did do so during t´he lockdowns. Thus the scale of the effects are rather different.
4) Actually no war was declared on Ukraine by Russia in February 2022, which is why they call it a "Special Military Operation". Similar to the Americans in Afghanistan, Iraq, Libya, Syria and so forth. If they did so, they would be legally obliged to laws of war.
There is a need to rethink the selection of cases used as its current form prevents an accurate and reliable scientific result, and the promises made in the abstract are not able to be fulfilled.
Author Response
We start by thanking the reviewer for the kind feedback and the valuable comments. Responses to the specific points follow.
- First and foremost, we further specified the viewpoint of the paper and its scope (lines 41-44 and 156-159). Also, we added some more details about each emergency throughout the introduction section of the paper (cf. point 3.)
- We updated the previous data and added new ones related to IFR (lines 62-68)
- We added several additional informations and literature references in order to describe and distinguish the effects of the two emergencies in more detail (lines 74-101)
- We applied the substitution in the formulation (lines 418 and 422) and, in some parts of the text, changed the term "war" to "conflict"
- We have now specified the selection criteria for the two cases at the start of the paper (lines 39-50), so as to make the terms of comparison more clear and thus the argument that follows
Given the amount of insertions added in the Introduction, we considered dividing it into three parts in order to make it easier to read: a brief general introduction of the object of the paper, a sub-section on the two selected emergency cases, and a sub-section on the research proposal.
Reviewer 3 Report
This is an interesting article in which the authors present a tool (Hyperion Observatory) for analyzing human interactions (social cohesion) during emergencies and also the results obtained using this tool in the context of two emergencies: the COVID-19 pandemic and the Russian-Ukrainian war.
In my opinion, the article requires revisions/clarifications before being accepted for publication- please see below.
Major comments:
- The introduction should be supplemented with data on the relationship between social cohesion and people's health during emergencies, in order to outline the theoretical framework of the research presented in the article.
- From the research presented, it is not clear the connection between social cohesion during emergencies and people's health, nor the way in which this connection can be evaluated based on the data collected by the Hyperion Observatory.
- The discussions should be revised, so that the research results are analysed and discussed in the context of the literature data.
Minor comments:
- Line 28 - please delete “McCann, 2022”.
- Line 30 - please replace “approx.” with “approximately”.
- Line 62 - please use square brackets instead of round brackets for bibliographic citation.
- Line 64 - a bibliographical citation should be added at the end of the paragraph.
- Line 119- the name of the community is blinded, although it is mentioned in the abstract - see L12- “Veneto community”.
- Figure 1 is not clear, it is very difficult to visualize. Moreover, the authors should explain, even briefly, the data presented in this figure.
Author Response
We start by thanking the reviewer for the kind feedback and the valuable comments. Responses to the specific points follow.
Major comments:
- The introduction should be supplemented with data on the relationship between social cohesion and people's health during emergencies, in order to outline the theoretical framework of the research presented in the article.
We added some additional informations and literature references (lines 104-107 and 110-118) in order to describe the mentioned-relationship and more precisely ground the research (see also the following point).
- From the research presented, it is not clear the connection between social cohesion during emergencies and people's health, nor the way in which this connection can be evaluated based on the data collected by the Hyperion Observatory.
We further detailed the connection between cohesion and people's health in emergency-cases, and how this is assessed by the Hyperion observatory, also using dedicated literature references (lines 143-159 and 163-166).
- The discussions should be revised, so that the research results are analysed and discussed in the context of the literature data.
We revised the Discussions retrieving literature references cited in the Introduction (where additional ones have been added) and with newly made additions (see that section in the attachment). We performed the same operation in the Conclusion section as well.
Minor comments:
- Line 28 - please delete “McCann, 2022”.
Deleted.
- Line 30 - please replace “approx.” with “approximately”.
Replaced.
- Line 62 - please use square brackets instead of round brackets for bibliographic citation.
Changed.
- Line 64 - a bibliographical citation should be added at the end of the paragraph.
Added (now line 120)
- Line 119- the name of the community is blinded, although it is mentioned in the abstract - see L12- “Veneto community”.
Fixed (throughout the paper).
- Figure 1 is not clear, it is very difficult to visualize. Moreover, the authors should explain, even briefly, the data presented in this figure.
We replaced Figure 1 with two separate, higher resolution images. The data have been explained before the new images (lines 345-347)
Round 2
Reviewer 3 Report
I thank the authors for the effort to respond to my comments and suggestions.
They adequately addressed all the suggestions and comments I made during the first evaluation of the article.
Therefore, I consider that the article can be accepted for publication in its current form.